REGISTERED REPORT PROTOCOL

# Effects of a locally available dietary interventions counselling on the community-based management of anaemia in children under five years in Ghana: Kumbungu cluster randomized controlled trial protocol

Benjamin Demah Nuertey[1,2]*, Alfred E. Yawson[1], Joyce A. Addai[3], Richard B. Biritwum[1]

**1** Community Health Department, University of Ghana Medical School, Korle-Bu, Accra, Ghana, **2** Public Health Department, Tamale Teaching Hospital, Tamale, Ghana, **3** Department of Medicine, Korle-Bu Teaching Hospital, Korle-Bu, Accra, Ghana

* ben.nuertey@gmail.com

## Abstract

Anaemia in children under five years remains a significant cause of mortality and morbidity in low-middle income countries. Globally, 27% of the world's population is anaemic, of which developing countries account for more than 89%. The global prevalence is worse in Africa and Asia. Anaemia has the potential of maintaining the cycle of poverty, as it prevents children from attaining their full development potential. An important part of anaemia in children under-five years are preventable. Locally available dietary (LAD) interventions may be the sustainable interventions to address the high prevalence of anaemia in our communities. The aim of the study is to determine the effect of counselling on Locally Available Diet, on anaemia among children aged 6–59 months in the Kumbungu District of the Northern Region, Ghana. This study will be a community-based cluster randomized, controlled trial, with two parallel arms; Iron + Folic Acid (IFA arm) hereby referred to as the Standard care arm and Iron + Folic Acid + Counselling on Locally available dietary intervention (IFA+LAD arm) referred to as the Standard Plus arm. Study participants are children between the ages of 6 to 56 months. All study participants would receive iron plus folic acid. The minimum number of children per arm is 330 and the number of community clusters is 10 (5 per study arm). Also, considering this study proposes two parallel arms, the total minimum study sample size of children under five years is 496, the minimum total number of community clusters is 10 and a minimum of 25 households per community cluster, 124 households per study arm and 248 households for the study. Randomization is at the level of the clusters (selected communities). The intervention group receives counselling on LAD at a household level aimed at promoting the intake of locally available iron, folate and vitamin B12 rich foods. Also encourage the intake of food rich in enhancers of iron absorption and discouraging the intake of food rich in inhibitors of iron absorption. The primary outcome is mean haemoglobin levels in study arms. Secondary outcomes would include measurement of weight, height/length, mid upper arm circumference, dietary record, serum iron, ferritin, and other

**Data Availability Statement:** No datasets were generated or analysed during the current study. All relevant data from this study will be made available upon study completion.

**Funding:** BDN received Part funding for this project from the Ghana-Michigan Collaborative funding program, which administers the Hauslohner Award. The amount was to take care of cost of reagents for the study. The principal investigator for this study would fund the remaining amount.

**Competing interests:** NO authors have competing interests

parameters of full blood count. Ethical clearance has been obtained and trial registered with Pan African Trial Registry (www.pactr.org) **PACTR201906918438423**.

## Introduction

Anaemia in children under five years is an important public health problem requiring urgent attention. Globally, 27% of the world's population is anaemic of which developing countries account for more than 89% [1,2]. Anaemia is among the top five causes of years lived with disability in the year 2016 [1]. Preschool children and pregnant women are the most affected. Globally, the prevalence of anaemia (defined as haemoglobin level of $<110$ g/L) in children aged 6–59 months is 43% [3]. The global prevalence is worse in Africa and Asia. In Africa, the prevalence of anaemia among children under the age of five years is estimated at 62% [4]. This is above the 40% cut off limit of the World Health Organization's (WHO) classification of anaemia as a severe public health problem [5]. The prevalence of anaemia in children under five years is estimated at 71% for west Africa [6]. In Ghana, the overall prevalence of anaemia in children under five years is 78.4% [7]. There are significant regional and community variations. In the northern region of Ghana, almost nine out of ten children under five years have anaemia [7]. A recent survey found contextual causes such as inadequate nutritional knowledge, food, diarrhoea, malaria, worm infestations, and inadequate dietary intake [8].

Anaemia has the potential of maintaining the cycle of poverty as it prevents children from attaining their full development potential [9]. These effects are often irreversible even if anaemia is corrected later in life [10]. In the shorterm, anaemia in children under five impairs cognitive, emotional, physical and brain development [11]. In the medium term, it negatively impacts on educational attainment and in the long-term, impacts negatively on adult life and earning abilities thereby increasing the likelihood of having anaemic children and maintaining the cycles of poverty and anaemia. Mortality estimates showed that, 5.8 million children under five years of age died in 2015 [11,12]. Sub-Saharan Africa in 2013 contributed 25% of global births and 50% of global deaths of children under five years which are respectively projected to reach 33% births and 60% deaths by 2030 [13]. Anaemia contributes significantly to deaths within this age group either directly or indirectly by complicating other conditions such as malaria, malnutrition, diarrhoea and pneumonia. It has been estimated that for each 1g/dl increase in hemoglobin, the risk of death falls by 24% [14]. Also 1.8 million deaths in children under five years within Africa could be avoided by increasing the hemoglobin of these children by 1g/dl.

Most of the causes of anaemia in children under-five years are preventable. Globally, Iron deficiency anaemia is the dominant causes of anaemia accounting for more than 60% of all anaemia [15]. Iron deficiency anaemia in children within developing countries is partly due to increased physiological demands associated with child growth [16] and also, reduced intake complicated by predominantly cereal based diet with high content of phytates, phenols and other ligands that impair iron absorption [17]. Increased loss of blood from hookworm infestation has also been estimated to account for a significant proportion of iron deficiency anaemia [18].

Reduction of the burden of anaemia requires multifaceted approaches, which have been a challenge for many developing countries resulting in over-reliance on hospital, based emergency management of anaemia where children often report too late to be saved. Cost implications for developing countries prevents adherence to universal iron supplementation

recommended by the WHO for areas with anaemia prevalence exceeding 40% [5]. Locally available dietary (LAD) interventions may be the sustainable interventions to address the high prevalence of anaemia in our communities. LAD interventions use available and abundant diets in the community to manage or prevent dietary problems. The use of the LAD may provide the cheapest and easily accessible and culturally appropriate options to manage anaemia. Almost all communities have some distinctive varieties of food with good nutritional values that with the right knowledge can be a tool to address health problems [19]. The aim of the study is **t**o determine the effect of counselling on Locally Available Diet, on anaemia among children aged 6–59 months in the Kumbungu District of the Northern Region, Ghana. The study would specifically determine the baseline prevalence of anaemia and iron deficiency anaemia among children under five years in the Kumbungu district of northern region. The study will also determine the factors associated with anaemia among children under five years and would test the effects of counselling on Locally Available Diet on anaemia among children aged 6–59 months in the Kumbungu district of the northern region, Ghana.

## Methods

### Study site

The study would be conducted in the Kumbungu district of the Northern region. Kumbungu in the northern region was selected because, it has one of the highest prevalence of anaemia in children under five years. Several activities by Governmental and Non-Governmental Organizations (NGOs) at addressing the problem of anaemia have shown little improvement. The Northern region has 26 districts. Kumbungu district is one of the smallest districts within the northern region with a total landmass of 1599 Square Kilometres. Kumbungu has 39,341 inhabitants within 4,133 households and a sex ratio of 100.2 males per 100 females. Children constitute 48.4% of the population. 7,101 of the population of Kumbungu are aged 0 to 4 years [20]. Average household size in Kumbungu is 9.5. The total fertility rate, crude birth rate and general fertility rate for the district are respectively 3.6, 23.4 per 1000 and 103.9 births per 1000 women aged 15–49. The household structure is mainly that of extended family system accounting for 71.8% of the household population while nuclear family system accounts for 14.6%. According to the recent census [21], only 15.2% of the population is literate in English language and 63.8% of persons aged 3 years and older in the district have never attended school. With regards to employment status, 98% of the population aged 15 years and older are employed in the private informal sector. Majority of the working population are in farming related occupations. The district has 115 communities, most with a population below 500. Only two communities have population above 5000. With regards to housing, 79.4% of the inhabitants in the district live in thatch/palm/raffia roofed houses [21]. The district houses the only major community water treatment facility in the northern region.

Culturally, meat is reserved for festive occasion. Though livestock, poultry are kept by some on small scaled free-range level, the eggs and meat are often sold for cash. Majority of the people in the Kumbungu district are farmers who cultivate small fragmented parcel of lands [22]. Cattles are the most valuable livestock that are rarely sold or slaughtered except for extreme shock such as crop failure or famine [22]. Also, smaller livestock and poultry are often kept by most household which provides immediate cash to purchase food, pay for medical bills, source of protein rich diet and also given out as gift/ sacrifice. Previous studies in the district have found high intake of phytates and tea which are known inhibitors of iron absorption [23]. Locally available diets that contain high amount of vitamin C such as tamarind, ripe shea fruit and baobab are seasonal and when available, are not expensive to buy.

## Study design

This study will be a community-based cluster randomized, controlled trial, with two parallel arms. Eight communities shall be selected randomly from the list of communities as reported by the 2010 population and housing census, district analytical report of Kumbungu district [21]. The eight selected communities (clusters) would be randomly assigned to intervention and control groups. Four community randomly selected by balloting would be assigned to intervention group while the remaining four would be assigned to control group. Thus, cluster assignment is at the level of community. All eligible children within the household would be included in the study. The intervention clusters would receive counselling on locally available diets (LAD) for anaemia management. All study participants would receive iron plus folic acid. The study would follow the closed cohort of eligible households with children aged between 6 to 56 months at baseline for 12 weeks. The counselling on locally available diets would be administered at baseline and a month after. Primary outcome data would be collected at baseline (pre-intervention) and then at the end of 12 weeks (post intervention). Eligible children within the household are the unit for analysis and the clustering would be taken into account.

Cluster randomization was chosen for practical reasons and also to minimize contamination. Children of the same household eat from the same pot. Since the study assess counselling interventions targeting intake and avoidance of enhancers and inhibitors of iron absorption in a closely knit community system, it was more appropriate to randomize at the level of the community cluster, instead of individual or household randomization. Counselling on LAD intervention would be administered at household level. Households must have at least a child aged 6–56 months at baseline. There would be no matching between the study arms. And there would be no blinding because the specific intervention is a counselling on LAD intervention which makes blinding almost impossible at the participant level. Also, to minimize contamination, a community cluster randomization instead of household randomization would be used. Also selected communities on intervention would be selected to ensure that there are other communities who are not part of the study in between them.

## Study arms

There shall be two study arms; Iron + Folic Acid (IFA) arm hereby referred to as the Standard care arm and Iron + Folic Acid + Counselling on Locally available dietary intervention (IFA +LAD) arm referred to as the Standard Plus arm.

## Description of study population

Study participants are children aged 6 to 56 months at baseline. Children less than six months are expected to be exclusively breastfed according to prevailing guidelines. Counselling on LAD that promotes intake of enhancers of iron absorption and avoiding the intake of inhibitors of iron absorption would not be appropriate in the below 6-month age group. Participants must also be less than 56 months of age at baseline because the intervention is expected to last for three months and it is expected that, participant recruited must be below five years at post intervention. The mothers or caregivers of eligible children would receive the counselling on locally available diets.

All eligible children 6–56 months of age at baseline would be included in the study so as to answer the first two specific objectives of the study, which seek to determine the prevalence and factors associated with anaemia among children less than five years. To answer the third specific objective, which is the cluster randomized control trial on the effect of counselling on anaemia status, only children whose haemoglobin level at baseline is less than 11.0g/dl would be included. All children with haemoglobin level at base line above 11.0g/dl would be excluded.

## Inclusion and exclusion criteria

**Inclusion criteria at household level.**   At the household level, household inclusion to the study must satisfy the following:

i.  The household must have at least a child aged 6 to 56 months at baseline.

## Inclusion criteria at the participant level

i.  All children within the eligible household aged 6 to 56 months at baseline would be included to answer specific objectives one and two.

ii.  For specific objective three, only children with haemoglobin concentration measured at baseline less than 11.0 g/dl would be eligible for inclusion.

iii.  The child should be residing in the selected community for at least the past three months.

iv.  Must have a legal guardian capable of providing informed consent.

**Exclusion criteria.**   Eligible children with any of the following would be ex6cluded from the study:

i.  Current infective illness (example; respiratory infection, diarrhoea) with fever. Current infective illness would be assessed from self-reported and verification of available medical records.

ii.  Diagnosed case of any clinical haemoglobinopathy (eg, beta-thalassemia major, HbE-beta thalassemia, Sickle cell disease). This will be assessed based on self-report and available medical and laboratory records.

iii.  Received iron supplements or iron containing MMP in the previous month.

## Sample size calculation

The sample size was calculated to determine the minimum difference in outcome of management of anaemia between arms of the study. The following information was used:

1.  Estimated prevalence of anaemia at baseline is 82% from previous study [7], and the sample size is predicted on an estimated absolute reduction of at least 22% of the baseline prevalence.

2.  Estimated average cluster size is 55 children under five years and an estimated number of children under five years with anaemia in each community clusters is 45

3.  An estimated intra-cluster correlation (ICC) of 0.035

4.  Power of at least 80% (0.80)

5.  Significance level of 5% (0.05)

6.  A dropout rate of about 5% for the 12 weeks period, and

7.  A contamination effect of 5%

With the above considerations, and using a menu driven facility for sample size calculation in cluster randomized controlled trial [24] available in STATA (version 14.1, StataCorp,

Special Edition, College Station, Texas 77845 USA), the minimum sample size and number of community clusters per arm was calculated. The minimum number of children per arm is 248 and the number of community clusters per study arm is 5. Also, considering this study proposes two parallel arms, the total minimum study sample size of children under five years is 496 [of which an expected 82% (n = 407) would have haemoglobin levels less than 11.0g/dl]. Also, the minimum total number of community clusters is 10. From the most recent census, the average number of children under five years per household is 1.7, which is approximately 2. This means that, we need to recruit a minimum of 25 households per community cluster, 124 households per study arm and 248 households for the study.

## Sampling process

A two-stage sampling approach will be used as follows:

- All communities in the district with total population at the 2010 census above 500 would be eligible for selection. We found 20 out of 115 communities met this criterion.

- The map of the district would be divided into north and south and one randomly selected.

- The selected area (north or south) would be further divided into four quadrants. Two quadrants would be selected using simple random sampling technique of balloting without replacement. The first selected quadrant would have four communities selected by simple random sampling and assigned to intervention group. The second selected quadrant would have four communities selected by simple random sampling and assigned to control group.

- If the selected communities are not large enough to give the desired number of at least 248 households and 496 children under five years, a simple random technique would be employed to select additional communities within the intervention quadrant or control quadrant until the desired number of household and children are achieved.

- All children within the selected communities would be eligible for inclusion

## Randomization

Randomization is at the level of the clusters (selected communities). The first selected quadrant of the selected half of the district shall be the intervention quadrant. Two communities would be selected from the intervention quadrant to be the intervention communities. The second selected quadrant would be the control quadrant. Two communities would be selected from the control quadrant to be the control communities. There would be no blocking and stratification.

## Allocation concealment mechanism and blinding

The intervention is a counselling intervention, which makes concealment and blinding a near impossible task. This is because, at the individual participant level, the mother/ care giver of the child should be aware that she/he is being counselled on some locally available diet interventions to improve anaemia in his/her child. All eligible patients within the cluster would be included in the study. Independent laboratory staff that would analyse the samples would be kept blind to intervention assignment of the communities.

## Interventions

Intervention pertains to selected intervention communities. The clusters assigned to counselling on LAD at a household level would each receive targeted one-on-one counselling on LAD

**Table 1. Baseline and Outcome measures, intermediary factors and time points.**

| Primary and Secondary outcomes | | Baseline | Week | | End-line |
|---|---|---|---|---|---|
| | | Week 0 | 4 | 8 | Week 12 |
| **Primary outcome** | | | | | |
| 1 | Haemoglobin concentration | X | | | X |
| **Secondary Outcomes** | | | | | |
| 2 | **Anthropometry** | | | | |
| | Weight | X | | | X |
| | Length/Height | X | | | X |
| | Mid Upper Arm Circumference | X | | | X |
| 3 | **Biochemical outcome** | | | | |
| | Serum Ferritin | X | | | X |
| | Serum Iron concentration | X | | | X |
| | Serum transferrin saturation | X | | | X |
| | Serum Vitamin B12 level | X | | | X |
| | Serum folate | X | | | X |
| 4 | **Clinical Outcome** | | | | |
| | *General Examination* | | | | |
| | Oedema | X | | | X |
| | Jaundice | X | | | X |
| | Pallor | X | | | X |
| | Skin and hair changes | X | | | X |
| 5 | **Dietary recall and quantification** | | | | |
| | 24 hour repeated food record | X | | | X |
| | Enhancers of iron in diet | X | X | X | X |
| | Inhibitors of iron in diet | X | X | X | X |
| | Supplements used during the study | X | X | X | X |

intervention aimed at promoting the intake of locally available iron, folate and vitamin B12 rich foods, food rich in enhancers of iron absorption and discouraging the intake of food rich in inhibitors of iron absorption. The counselling would be administered at baseline and repeated monthly. Each session of counselling would last minimum of 20 minutes. The counselling guide is attached as supplement. The rest of the interventions are targeted at the individual participant level. All study participants would receive the current practiced standard treatment for anaemia; iron + folic acid.

The control or standard care arm would receive only the iron plus folic acid syrup. The standard plus arm (intervention arm) would receive iron, folic acid syrup treatment plus counselling on LAD intervention. All two arms of the study would run parallel and would be randomized in the ratio 1:1. The dosage of Iron and folic acid to be dispensed would be according to the current WHO recommended dose for age of the child [25]. The mothers/ legal caretakers would be advised to administer the supplement to children in the morning 1 to 2 hours before or after meals and not to be combined with meals.

Each study participant is required to take the prescribed doses daily. Monthly iron and folic acid requirement shall be dispensed to mothers/guardians. To address the issue of compliance to administered iron and folic acid syrups, a member of the study team shall visit each household every week to inspect the administration of the iron and folic acid syrup by observing the level of syrup in the bottles to ascertain compliance. A three-day food record would be administered at baseline and at end line. The food record question and a food frequency questionnaire are attached as supplement.

Counselling on LAD would be carried out by trained nurses with at least a diploma degree in nursing. A training on the counselling guide would be carried out by a team of nutritionist and doctor. The training is expected to last six hours and would include lecture, simulation and practical discussions. Food groups, locally available foods, enhancers of iron, inhibitors of iron, effective counselling methods would be discussed.

## Outcome measures

Outcome measures mainly pertain to individual participant level. The study will evaluate the effect of counselling on LAD interventions on anaemia status of children in the study arm.

**Primary outcome.** The primary outcome measure is a reduction in the prevalence of anaemia within the study arms. The study would measure haemoglobin of children using haematological analyser and classify children with Hb <11.0 g/dL as anaemic. Since only children with Hb <11.0 g/dL would be included in the intervention, the baseline prevalence of anaemia in each arm would be 100%. At end line, the prevalence of anaemia within study arm would be determined by estimating the absolute proportion of children whose haemoglobin concentration is <11.0 g/dL.

## Secondary outcome

Several secondary outcomes would be measured in this study. Secondary outcomes would include measurement of anthropometry; weight, height/length, mid upper arm circumference. Others would include dietary record; 24hour repeated food record, two on weekdays and one on a weekend at baseline and repeated at the end of the study. Blood and serum parameters such as serum iron, ferritin, and other parameters of full blood count would also be taken. Table 1 displays the baseline and outcome measures and the time points.

## Recruitment of study participants and procedure for data collection

The process of approaching and enlisting households will be based on the following guidelines:

i. Conduct a quick House and Household Listing in all four selected communities

ii. Survey eligible households with the selected village

iii. Interview household Heads/Spouse (Care givers) per each household

iv. All eligible children per selected household would be enumerated and included in the study

The data collection team would visit households and administer household questionnaires as well as individual child questionnaires after completing consent and assent procedures. A trained phlebotomist would carry out the bloodletting. Five millilitres of blood samples would be taken for haematological and biochemical analysis. The trained community health nurses would carry out the counselling intervention at week one, four and eight of the study. The independent data collection team would be blinded to the community randomisation. Other information to be collected would include household characteristics, medication history, details about intake of IFA supplements, adverse events of intervention, morbidity and mortality outcomes.

## Ethical consideration

Written informed consent would be sought at the household level from the head of the household. Verbal assent would be obtained from individual study participant. The consent would

be obtained at the baseline. Ethical clearance was obtained from the Tamale Teaching hospital institutional review board TTHERC/20/06/19/01. The trial has been registered with the Pan African Trial Registry (www.pactr.org) with unique identification number for the registry as **PACTR201906918438423**.

## Data analysis

Data analysis would be by intention to treat basis. Descriptive statistics would be used to describe and display baseline characteristics of study participants by study arms to allow for comparing of the study arm after randomisation. The primary outcomes are the change in mean haemoglobin concentration and change in proportion of iron deficiency calculated from serum iron results for each arm of the study. This would be obtained by computing the mean change in haemoglobin concentration of each study participant within the study arm. For comparing arms, we would use mixed effects models with random effects at the cluster level for all outcomes, but including baseline adjustment for any continuous outcomes. Thus, we will adapt the treatment effect analysis approach considering the design effect of the study. Analysis of variance (equality of means test statistic) would be used to test the mean haemoglobin between categorical variables. Change in proportion of anaemic (ie, haemoglobin concentration <110 g/L), severely anaemic (ie, haemoglobin concentration <70 g/L) would be obtained for all groups. Nutritional z-scores will be calculated according to the WHO growth reference curves with the use of Epi Info version 7.2.2.2. Weight-for-height, height-for-age, and weight-for-age z scores of less than −2 would be classified as wasting, stunting, and underweight, respectively; z scores of less than −3 would be considered to indicate severe wasting, or severe underweight. Underweight (ie, weight-for-age Z scores less than −2), and wasting (ie, weight-for-height Z scores less than −2) would be analysed. Multivariate data analysis would be adapted considering a priori selected confounders based on literature (age in months and sex).

An impact evaluation analysis would be adopted to establish the hypothetical impact of the intervention. Multilevel mixed-effects generalized linear model (MEGLM) would be adopted based on the design nature of the study. The assumption for using MEGLM would allow for conditional normally of random effects.

Using a handy measure conversion, handbook, the gram weight of the food items consumed would be obtained and entered into the West Africa Food Analysis software for the analysis of nutrient/dietary data.

## Discussion

The prevalence of anaemia in children under-five years is high and there is the need to reduce this burden. Iron supplementation programs have not achieved their full-anticipated effects because the bioavailability of non-haem iron is low. Many children continue to die because of anaemia and several others live with countless developmental disabilities. Given the abnormally high prevalence of anaemia in children under five years, and it's associated enormous public health implications of developmental disabilities across the life span [26], better methods are needed to improve iron supplementation and reduce the prevalence of anaemia within this age group. Also, recommended iron supplementation for the treatment of anaemia in children usually produce large increase in colonic iron because, typical iron absorption is less than 20% of dose ingested [27]. Non-absorbed iron could be harmful to children because of its role in modifying the gut microbiota with the resultant effect of increasing intestinal pathogens [28]. Increasing number of studies in children under five years reports adverse effects with iron supplementation such as decrease growth, increased diarrhoea, interaction with other

trace elements and increased inflammatory markers [29]. It is essential to explore new cost-effective ways of enhancing iron absorption and improving safety of supplemented iron. The possible synergistic role of these LAD activities outlined with IFA supplementation has not been fully explored. Individual anaemic children admitted or managed at health centres and or hospital might have been given supplemental iron, folate and or blood transfusion. However, there is no program in Ghana for iron and folate supplementation for children under 5 years. It is the hope that information from the study would be helpful in programmatic decision making.

## Steps to addressing known confounders and limitations

Beyond the process of randomization, the following steps would be taken to address known confounders that may impact on the quality of the interventions. Intestinal worms can impair iron absorption or cause loss of iron leading to or worsening anaemia. Hence all children in the study would be given suspension Albendazole at baseline. Also, malaria parasite can cause red blood cell haemolysis and anaemia. Hence all eligible children would be tested for malaria and those found to have malaria referred for treatment. Also, the exclusion criteria are designed to exclude other potential confounders such as children with known hemoglobinopathies. Household food insecurity may be a confounder that may influence adoption of counselling intervention promoting intake of enhancers of iron absorption and discouraging the intake of inhibitors of iron absorption. Hence there is the need to measure at baseline and end-line food access.

## Supporting information

**S1 Text. Kumbungu cluster randomized controlled trial protocol.**
(DOCX)

**S2 Text. SPIRIT checklist.**
(DOCX)

**S1 File.**
(DOCX)

## Author Contributions

**Conceptualization:** Benjamin Demah Nuertey, Alfred E. Yawson, Joyce A. Addai, Richard B. Biritwum.

**Funding acquisition:** Benjamin Demah Nuertey, Alfred E. Yawson.

**Methodology:** Benjamin Demah Nuertey, Alfred E. Yawson, Joyce A. Addai, Richard B. Biritwum.

**Project administration:** Benjamin Demah Nuertey.

**Resources:** Benjamin Demah Nuertey, Joyce A. Addai.

**Supervision:** Benjamin Demah Nuertey, Alfred E. Yawson, Richard B. Biritwum.

**Writing – original draft:** Benjamin Demah Nuertey.

**Writing – review & editing:** Benjamin Demah Nuertey, Alfred E. Yawson, Joyce A. Addai, Richard B. Biritwum.

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
