## [Decision Letter · Decision Letter 0]

21 Jan 2021

PONE-D-20-28388

Effects of a locally available dietary interventions counselling on the community-based management of anaemia in children under five in Ghana: Kumbungu cluster randomized controlled trial protocol

PLOS ONE

Dear Dr. Nuertey,

Thank you for submitting your manuscript to PLOS ONE. After careful consideration, we feel that it has merit but does not fully meet PLOS ONE’s publication criteria as it currently stands. Therefore, we invite you to submit a revised version of the manuscript that addresses the points raised during the review process.

We look forward to receiving your revised manuscript.

Kind regards,

Seth Adu-Afarwuah

Academic Editor

PLOS ONE

Journal Requirements:

2.We note that Figure(s) 1 in your submission contain map images which may be copyrighted. All PLOS content is published under the Creative Commons Attribution License (CC BY 4.0), which means that the manuscript, images, and Supporting Information files will be freely available online, and any third party is permitted to access, download, copy, distribute, and use these materials in any way, even commercially, with proper attribution. For these reasons, we cannot publish previously copyrighted maps or satellite images created using proprietary data, such as Google software (Google Maps, Street View, and Earth). For more information, see our copyright guidelines: http://journals.plos.org/plosone/s/licenses-and-copyright.

a)   You may seek permission from the original copyright holder of Figure(s) 1 to publish the content specifically under the CC BY 4.0 license. 

3. Please include a caption for figure 1.

4. We note you have included a table to which you do not refer in the text of your manuscript. Please ensure that you refer to Table 1 in your text; if accepted, production will need this reference to link the reader to the Table.

Reviewers' comments:

Reviewer's Responses to Questions

**Comments to the Author**

1. Does the manuscript provide a valid rationale for the proposed study, with clearly identified and justified research questions?

Reviewer #1: Yes

Reviewer #2: Yes

Reviewer #3: Yes

2. Is the protocol technically sound and planned in a manner that will lead to a meaningful outcome and allow testing the stated hypotheses?

Reviewer #1: Yes

Reviewer #2: Yes

Reviewer #3: Partly

3. Is the methodology feasible and described in sufficient detail to allow the work to be replicable?

Reviewer #1: Yes

Reviewer #2: Yes

Reviewer #3: No

4. Have the authors described where all data underlying the findings will be made available when the study is complete?

Reviewer #1: Yes

Reviewer #2: Yes

Reviewer #3: Yes

5. Is the manuscript presented in an intelligible fashion and written in standard English?

Reviewer #1: Yes

Reviewer #2: Yes

Reviewer #3: No

6. Review Comments to the Author

You may also provide optional suggestions and comments to authors that they might find helpful in planning their study.

Reviewer #1: The authors describe testing a promising intervention for the prevention and treatment of anemia in young children in Africa.

The study design appears sound and methods are generally adequately described.

English errors occur throughout the manuscript and the figures and tables contain alignment and formatting errors.

Comments:

Please describe the diet of the communities and whether the residents have access to the foods/beverages you plan to include in the counseling intervention. Is there published evidence that consumption of iron inhibitors and lack of consumption of iron enhancers is occurring in these communities? Are animal foods commonly consumed?

How will nutrient/dietary data be analyzed? What databases are to be used?

Is the sample size adequate to detect significant differences in the secondary outcome measures of s ferritin and iron?

Have these communities been exposed to supplemental iron and folate previously?

What instructions are provided to the caregivers regarding administration of the supplements (with meals or not; morning or evening)?

Explain the training for the LAD counseling.

The consent form should describe the collection of dietary data.

Reviewer #2: For the secondary outcomes, diet recall is often not accurate. A better approach would be to have the participants record food intake daily for more accuracy. Also information on supplements used during the study, if any, by the households should be included.

Reviewer #3: Abstract

- since the cluster is the unit of randomisation, indicate the total number of children, households and community clusters to be included.

- clarify which aspects of anthropometry will be measured.

Introduction

- typo in second-last paragraph of page 2 - 'diarrhoea'

- in first paragraph on page 3, you mean 'children under five years [of age]' not 'children lower than five years'. Also, 'which are respectively projected...' instead of 'is respectively projected...'

- second-last line on page 3 - you probably mean 'distinctive' rather than 'peculiar'.

Methods

- there's a potentially fundamental problem with the study design. In the design section, you refer to the communities as the clusters. So your effective sample size is two clusters per arm; however in the sample size calculation, you refer to the households as the clusters, and indicate that the number of clusters per study arm is 54. This needs to be clarified. Either the community is a higher-level unit of stratification (in the same way one might conduct stratified randomisation by country in a multi-country study), or if it is indeed the unit of clustering then the study design and sample size calculation needs to be thought through, as it is not appropriate if cluster randomisation is at the community level.

- a more appropriate analysis would involve a mixed effects generalised linear model of the endline outcome measures adjusted for baseline measures (for continuous outcomes only - analysis of covariance), with further adjustment for age, sex and any stratification factors. Mixed effects models, or other hierarchical models, appropriately adjust for the clustering in the expected data. The proposed approach involving a t-test is not appropriate as it doesn't adjust for the clustered design, and doesn't allow for adjustment for baseline continuous outcomes (which improves efficiency - for continuous outcomes only).

7. PLOS authors have the option to publish the peer review history of their article (what does this mean?). If published, this will include your full peer review and any attached files.

Reviewer #1: No

Reviewer #2: **Yes: **Colleen M Croniger

Reviewer #3: No

---

## [Author Response · Author response to Decision Letter 0]

7 Jul 2021

Response to Reviewers’ comment

Thank you for the opportunity to submit a revision to this manuscript:

PONE-D-20-28388

Effects of a locally available dietary interventions counselling on the community-based management of anaemia in children under five in Ghana: Kumbungu cluster randomized controlled trial protocol

I am grateful for the inputs made by you and the reviewers, which greatly improved this manuscript.

Please find below the point-by-point response to reviewer comments.

Comments by Academic Editor:

2. We note that Figure(s) 1 in your submission contain map images which may be copyrighted …….. If you are unable to obtain permission from the original copyright holder to publish these figures under the CC BY 4.0 license or if the copyright holder’s requirements are incompatible with the CC BY 4.0 license, please either i) remove the figure or ii) supply a replacement figure that complies with the CC BY 4.0 license. Please check copyright information on all replacement figures and update the figure caption with source information. If applicable, please specify in the figure caption text when a figure is similar but not identical to the original image and is therefore for illustrative purposes only.

3. Please include a caption for figure 1

Response:

Thank you, the figure has been removed

4. We note you have included a table to which you do not refer in the text of your manuscript. Please ensure that you refer to Table 1 in your text; if accepted, production will need this reference to link the reader to the Table.

Response

Please this has been carried out

Response

Please this has been carried out

Reviewer #1: 

The authors describe testing a promising intervention for the prevention and treatment of anemia in young children in Africa.

The study design appears sound and methods are generally adequately described.

English errors occur throughout the manuscript and the figures and tables contain alignment and formatting errors.

Response 

Thank you, English Errors corrected and the figure and table alignment and formatting errors corrected

Comments:

Please describe the diet of the communities and whether the residents have access to the foods/beverages you plan to include in the counseling intervention. Is there published evidence that consumption of iron inhibitors and lack of consumption of iron enhancers is occurring in these communities? 

response

Majority of the people in the Kumbungu district are farmers who cultivate small fragmented parcel of lands [22]. Cattles are the most valuable livestock that are rarely sold or slaughtered except for extreme shock such as crop failure or famine [22]. Also, livestock and poultry are often kept by most household which provides immediate cash to purchase food, pay for medical bills, source of protein rich diet and also given out as gift/ sacrifice [23]. Previous studies in the district have found high intake of phytates and tea which are known inhibitors of iron absorption [23]. Locally available diets that contain high amount of vitamin C such as tamarind, ripe shea fruit and baobab are seasonal and when available, are not expensive to buy.

comment

Are animal foods commonly consumed?

Response 

The following has been included on the section on study site; “Culturally, meet is reserved for festive occasion. Animal food; anyone who can afford animal food is a plus. Though livestock, poultry are kept by some on small scale, free range level, the eggs and meat are often sold for cash.”

comment

How will nutrient/dietary data be analyzed? What databases are to be used?

Response 

Using a handy measure conversion, handbook, the gram weight of the food items consumed would be obtained and entered into the West Africa Food Analysis software for the analysis of nutrient/dietary data. The software makes use of the West African Food composition table.

This has been included in the data analysis section.

Comment

Is the sample size adequate to detect significant differences in the secondary outcome measures of s ferritin and iron?

Response 

Thank you for pointing this to us. The sample size has been recalculated and powered to detect secondary outcome measures of serum ferritin and iron. Eight community clusters would be included and a minimum of 200 households with a minimum of 396 children would be included in the study.

comment

Have these communities been exposed to supplemental iron and folate previously?

Response 

Individual anaemic children admitted or managed at health centres and or hospital might have been given supplemental iron and folate. However, there is no program in Ghana for iron and folate supplementation for children under 5 years. This statement was included in the discussion section.

Comment

What instructions are provided to the caregivers regarding administration of the supplements (with meals or not; morning or evening)?

Response

Women would be advised to administer the supplement to children in the morning 1 to 2 hours before or after meals and not to be combined with meals.

comment

Explain the training for the LAD counseling.

Response

Thank you for pointing this out. The following has been included in the section on intervention; last paragraph. 

“Counselling on LAD would be carried out by trained nurses with at least a diploma degree in nursing. A training on the counselling guide would be carried out by a team of nutritionist and doctor. The training is expected to last six hours and would include lecture, simulation and practical discussions. Food groups, locally available foods, enhancers of iron, inhibitors of iron, effective counselling methods would be discussed.”

comment

The consent form should describe the collection of dietary data.

Response 

Thank you. It has been included.

Reviewer #2: 

For the secondary outcomes, diet recall is often not accurate. A better approach would be to have the participants record food intake daily for more accuracy. 

Response 

Thank you for emphasizing this. We would take a 24hour repeated food record. Two on weekdays and one on a weekend at baseline and repeated at the end of the study.

Comment

Also information on supplements used during the study, if any, by the households should be included.

Response 

Thank you very much. Supplements used during the study if any is a potential confounder and information on that would be collected. This has been included in the table 1 and the preceding text on secondary outcomes to me measured

Reviewer #3: 

Abstract

- since the cluster is the unit of randomisation, indicate the total number of children, households and community clusters to be included.

Response 

Thank you, this has been clearly stated now as can be found in the section under sample size calculation and the abstract and restated here as follows: “The minimum number of children per arm is 198 and the number of community clusters is 8 (4 per study arm). Also, considering this study proposes two parallel arms, the total minimum study sample size of children under five with anaemia is 396 and the minimum total number of community clusters is 8 and a minimum of 25 households per community cluster, 100 households per study arm and 200 households need to be enrolled for the study”.

Comment

- clarify which aspects of anthropometry will be measured.

response

Weight, height/length, mid upper arm circumference was specified in the abstract section now.

Comment

Introduction

- typo in second-last paragraph of page 2 - 'diarrhoea'

- in first paragraph on page 3, you mean 'children under five years [of age]' not 'children lower than five years'. Also, 'which are respectively projected...' instead of 'is respectively projected...'

- second-last line on page 3 - you probably mean 'distinctive' rather than 'peculiar'.

Response 

Really appreciate the suggestions; all were accepted and the changes effected.

comment

Methods

- there's a potentially fundamental problem with the study design. In the design section, you refer to the communities as the clusters. So your effective sample size is two clusters per arm; however in the sample size calculation, you refer to the households as the clusters, and indicate that the number of clusters per study arm is 54. This needs to be clarified. Either the community is a higher-level unit of stratification (in the same way one might conduct stratified randomisation by country in a multi-country study), or if it is indeed the unit of clustering then the study design and sample size calculation needs to be thought through, as it is not appropriate if cluster randomisation is at the community level.

Response 

Thank you very much for pointing this out to us. We worked out the whole sample size calculation and design section accepting your useful recommendation. Community clusters were used instead of household clusters and the rational explained in the design section. In all, eight community clusters were to be included in the study after re-calculating the sample size to accommodate your suggestion. The design and sample size sections have been reviewed to reflect the changes.

Comment

- a more appropriate analysis would involve a mixed effects generalised linear model of the endline outcome measures adjusted for baseline measures (for continuous outcomes only - analysis of covariance), with further adjustment for age, sex and any stratification factors. Mixed effects models, or other hierarchical models, appropriately adjust for the clustering in the expected data. The proposed approach involving a t-test is not appropriate as it doesn't adjust for the clustered design, and doesn't allow for adjustment for baseline continuous outcomes (which improves efficiency - for continuous outcomes only).

Response 

We agree with reviewer and have reviewed our initial analysis plan as below;

 An impact evaluation analysis would be adopted to establish the hypothetical impact of the intervention. Multilevel mixed-effects generalized linear model (MEGLM) would be adopted based on the design nature of the study. The assumption for using MEGLM would allow for conditional normally of random effects. Very helpful.

Thank you

Benjamin Nuertey

---

## [Decision Letter · Decision Letter 1]

7 Dec 2021

PONE-D-20-28388R1

Effects of a locally available dietary interventions counselling on the community-based management of anaemia in children under five in Ghana: Kumbungu cluster randomized controlled trial protocol

PLOS ONE

Dear Dr. Benjamin Demah Nuertey,

Thank you for submitting your manuscript to PLOS ONE. After careful consideration, we feel that it has merit but does not fully meet PLOS ONE’s publication criteria as it currently stands. Therefore, we invite you to submit a revised version of the manuscript that addresses the points raised during the review process.

Please address all reviewers' comments, and in particular please pay attention to the requests for revisions of your statistical analyses.

We look forward to receiving your revised manuscript.

Sincerely,

Yann Benetreau, PhD

Senior Editor, *PLOS ONE*

Reviewers' comments:

Reviewer's Responses to Questions

**Comments to the Author**

1. Does the manuscript provide a valid rationale for the proposed study, with clearly identified and justified research questions?

Reviewer #1: Yes

Reviewer #3: Yes

2. Is the protocol technically sound and planned in a manner that will lead to a meaningful outcome and allow testing the stated hypotheses?

Reviewer #1: Yes

Reviewer #3: Yes

3. Is the methodology feasible and described in sufficient detail to allow the work to be replicable?

Reviewer #1: Yes

Reviewer #3: Yes

4. Have the authors described where all data underlying the findings will be made available when the study is complete?

Reviewer #1: No

Reviewer #3: Yes

5. Is the manuscript presented in an intelligible fashion and written in standard English?

Reviewer #1: No

Reviewer #3: Yes

6. Review Comments to the Author

You may also provide optional suggestions and comments to authors that they might find helpful in planning their study.

Reviewer #1: The authors have made most of the recommended revisions and the manuscript is improved.

English errors remain and need correction. "Years" should be included when describing ages of children over 5 and adults.

Include articles (such as "the") when appropriate and check agreement of nouns and verbs for singular/plural.

"Meet" should be spelled "meat."

Reviewer #3: Thank you for your clear and comprehensive responses to the previous round of reviews. There are a number of key aspects of the protocol which I think would still benefit from improvement.

First, I would suggest that you revisit your sample size calculation, as this aspect of the study design is critical but doesn't seem correct in the current version of the protocol.

The key parameters of the sample size calculation are that the prevalence of anaemia is 82% and you hope to see an absolute 22% reduction over 12 weeks. You hope to have 80% power to detect this difference, at the 5% level of significance.

The cluster design aspects of the trial are inter-related. The cluster size and number of clusters to include are within your control. However, the ICC (ie. how similar the outcome is between individuals in the same community) is a natural phenomenon which you cannot vary. The design effect is the inflation factor by which you increase the basic sample size to account for the clustered design; it is function of the cluster size, number of clusters and ICC and does not need to be presented/calculated separately as you have done in (7) in your description of the calculation.

Given that the cluster size and number of clusters is what is within your control, these are the only aspects of the design that you can change. Currently you have decided that you'll recruit about 50 children per cluster - this is the cluster size. With this cluster size, assuming an ICC of 0.01, you will need 6 clusters per arm - so your current estimate of 4 per arm would not be sufficient.

If you are really only able to work with 4 communities in each arm, then you need to increase the cluster size i.e. the number of children you recruit in each cluster. With a cluster size of 100, all else remaining the same, you need 4 community clusters per arm, i.e. 8 community clusters total and 800 children total. If you anticipate any losses to follow-up, contamination/cross-over e.t.c. then you need to appropriately inflate the expected number of children per arm so that your actual cluster size will be more than 100 and the total expected number of children across the 8 community clusters will be more than 800.

Secondly, your analysis section still refers to the use of paired t-tests to compare continuous outcomes - this will not be appropriate for a clustered design, since the ordinary t-test does not include adjustment for clustering. For comparing arms, please plan to use mixed effects models with random effects at the cluster level for all outcomes, but including baseline adjustment for any continuous outcomes. The analysis section also says the study is powered to detect differences in some z-scores, yet the sample size calculation is only based on a reduction in prevalence of anaemia.

Lastly, you describe stepwise variable selection for variables to be included in the adjusted models - this is not appropriate for a randomised trial. In a randomised trial you only adjust for a priori potential confouners (e.g. age and sex) and any variables used for stratification - you don't use automated processes to identify variables to adjust for because by design you don't expect characteristics to differ between the randomised groups; if you expect any differences they you stratify randomisation by those characteristics, and then you also adjust for the stratification variables.

Please review these issues and if needed consult a statistician with experience in design and analysis of cluster randomised trials for further advice.

7. PLOS authors have the option to publish the peer review history of their article (what does this mean?). If published, this will include your full peer review and any attached files.

Reviewer #1: No

Reviewer #3: No

---

## [Author Response · Author response to Decision Letter 1]

12 Dec 2021

Reviewer #1:

Comment

The authors have made most of the recommended revisions and the manuscript is improved.

English errors remain and need correction. "Years" should be included when describing ages of children over 5 and adults. Include articles (such as "the") when appropriate and check agreement of nouns and verbs for singular/plural. "Meet" should be spelled "meat."

Response

Thank you very much for your thorough review which improved the manuscript. All the suggested changes have been carried out. Thank you.

Reviewer #3: 

Comment

Thank you for your clear and comprehensive responses to the previous round of reviews. There are a number of key aspects of the protocol which I think would still benefit from improvement.

Response

We are deeply grateful for your suggestions, education, and support particular with the statistical aspect of this protocol. Your inputs were very valuable to us. Thank you.

Comment

First, I would suggest that you revisit your sample size calculation, as this aspect of the study design is critical but doesn't seem correct in the current version of the protocol.

The key parameters of the sample size calculation are that the prevalence of anaemia is 82% and you hope to see an absolute 22% reduction over 12 weeks. You hope to have 80% power to detect this difference, at the 5% level of significance.

The cluster design aspects of the trial are inter-related. The cluster size and number of clusters to include are within your control. However, the ICC (ie. how similar the outcome is between individuals in the same community) is a natural phenomenon which you cannot vary. The design effect is the inflation factor by which you increase the basic sample size to account for the clustered design; it is function of the cluster size, number of clusters and ICC and does not need to be presented/calculated separately as you have done in (7) in your description of the calculation.

Given that the cluster size and number of clusters is what is within your control, these are the only aspects of the design that you can change. Currently you have decided that you'll recruit about 50 children per cluster - this is the cluster size. With this cluster size, assuming an ICC of 0.01, you will need 6 clusters per arm - so your current estimate of 4 per arm would not be sufficient.

If you are really only able to work with 4 communities in each arm, then you need to increase the cluster size i.e. the number of children you recruit in each cluster. With a cluster size of 100, all else remaining the same, you need 4 community clusters per arm, i.e. 8 community clusters total and 800 children total. If you anticipate any losses to follow-up, contamination/cross-over e.t.c. then you need to appropriately inflate the expected number of children per arm so that your actual cluster size will be more than 100 and the total expected number of children across the 8 community clusters will be more than 800.

Response

Thank you for taking your time to give an in-depth education on cluster sample size calculations. We followed your advice and your suggested reviews were carried out. First we dropped the calculation on design effect as suggested. Also, we inputted the estimated average number of children per cluster which currently stands at 75 in to the calculation, maintaining all other variables as suggested. This gave us a total of four clusters per study arm and a total of 8 clusters. 

Considering a 10% for loss to follow-up and contamination, our final sample size stands at 660 children under five years. Thus 330 children per arm of the study. This was possible with your kind suggestion. Thank you!

Comment

Secondly, your analysis section still refers to the use of paired t-tests to compare continuous outcomes - this will not be appropriate for a clustered design, since the ordinary t-test does not include adjustment for clustering. For comparing arms, please plan to use mixed effects models with random effects at the cluster level for all outcomes, but including baseline adjustment for any continuous outcomes. The analysis section also says the study is powered to detect differences in some z-scores, yet the sample size calculation is only based on a reduction in prevalence of anaemia.

Response

Thank you immensely for pointing out this to us. We have modified the data analysis section, (line 6). Hence, for comparing arms, we would use mixed effects models with random effects at the cluster level for all outcomes, but including baseline adjustment for any continuous outcomes. Thus, we will adapt the treatment effect analysis approach considering the design effect of the study.

The aspect that referred to the following statement was expunged; “The analysis section also says the study is powered to detect differences in some z-scores, yet the sample size calculation is only based on a reduction in prevalence of anaemia.” 

Comment

Lastly, you describe stepwise variable selection for variables to be included in the adjusted models - this is not appropriate for a randomised trial. In a randomised trial you only adjust for a priori potential confounders (e.g. age and sex) and any variables used for stratification - you don't use automated processes to identify variables to adjust for because by design you don't expect characteristics to differ between the randomised groups; if you expect any differences they you stratify randomisation by those characteristics, and then you also adjust for the stratification variables. Please review these issues and if needed consult a statistician with experience in design and analysis of cluster randomised trials for further advice.

Response

Thank you for pointing out this to us. The aspect of the proposed analysis describing stepwise variable selection was deleted. Multivariate data analysis would be adapted considering a priori selected confounders based on literature (age in months and sex).

---

## [Decision Letter · Decision Letter 2]

8 Feb 2022

PONE-D-20-28388R2Effects of a locally available dietary interventions counselling on the community-based management of anaemia in children under five years in Ghana: Kumbungu cluster randomized controlled trial protocolPLOS ONE

Dear Dr. Nuertey,

Thank you for submitting your protocol to PLOS ONE. I was asked to step in as new editor, and after reading the protocol, I feel it has merit for publication in PLOS ONE’s after some minor revisions. Therefore, we invite you to submit a revised version of the manuscript that addresses the points raised by reviewer 3.

In addition, please note the results of the national micronutrient survey in Ghana, published by Wegmuller et al in 2020 in PlosOne, showing iron deficiency contributed to roughly 30% of the anemia observed in children only. Hence, in your abstract, the sentence "Most of the causes of anaemia in children under-five years are preventable." should be rephrased to 'an important part of anemia...."

We look forward to receiving your revised manuscript.

Kind regards,

Frank Wieringa, M.D., Ph.D.

Academic Editor

PLOS ONE

Journal Requirements:

Reviewers' comments:

Reviewer's Responses to Questions

**Comments to the Author**

1. Does the manuscript provide a valid rationale for the proposed study, with clearly identified and justified research questions?

Reviewer #1: Yes

Reviewer #3: Yes

2. Is the protocol technically sound and planned in a manner that will lead to a meaningful outcome and allow testing the stated hypotheses?

Reviewer #1: Yes

Reviewer #3: Yes

3. Is the methodology feasible and described in sufficient detail to allow the work to be replicable?

Reviewer #1: Yes

Reviewer #3: Yes

4. Have the authors described where all data underlying the findings will be made available when the study is complete?

Reviewer #1: Yes

Reviewer #3: Yes

5. Is the manuscript presented in an intelligible fashion and written in standard English?

Reviewer #1: Yes

Reviewer #3: Yes

6. Review Comments to the Author

You may also provide optional suggestions and comments to authors that they might find helpful in planning their study.

Reviewer #1: The authors have made the revisions suggested and I have no further recommendations. I consider it acceptable for publication.

Reviewer #3: The manuscript is much improved from the previous round of reviews and I commend the authors for taking previous comments on board. I have two main points to raise in this round:

(1) the sample size calculation only works if I assume a higher ICC of around 0.035, not 0.01 as cited in point 3 on page 8, and if I ignore the coefficient of variation of 0.9 cited in point 4, which is much too high given the parameters already specified. I would recommend that the authors drop point 4 as it is determined by other parameters, and also recheck the ICC used in the calculation. Another minor point is to be clear in point 1 on page 7 that the sample size is predicated on an estimated ABSOLUTE reduction of at least 22%

(2) given that the sample size calculation is based on the prevalence of anaemia, it is incorrect to state that the primary outcome is haemoglobin (page 10). The primary outcome should be reported as anaemia, as this is what the sample size calculation is based on. The authors should then go ahead and describe how anaemia will be determined, i.e. measuring haemoglobin using an analyser and classifying children with values below [threshold] as anaemic. Most of this is already indicated in page 10-11. Further, the authors should focus the primary outcome data analysis on anaemia and not haemoglobin (haemoglobin may be analysed as a secondary outcome). If the authors intended haemoglobin to be the primary outcome, then a whole new sample size calculation appropriate for this continuous outcome is required.

7. PLOS authors have the option to publish the peer review history of their article (what does this mean?). If published, this will include your full peer review and any attached files.

Reviewer #1: No

Reviewer #3: No

---

## [Author Response · Author response to Decision Letter 2]

14 Feb 2022

Dear Dr. Frank Wieringa

Thank you very much for agreeing to step in as new academic editor

Academic editor comment

Please note the results of the national micronutrient survey in Ghana, published by Wegmuller et al in 2020 in PlosOne, showing iron deficiency contributed to roughly 30% of the anemia observed in children only. Hence, in your abstract, the sentence "Most of the causes of anaemia in children under-five years are preventable." should be rephrased to 'an important part of anemia...."

Response

Thank you, suggestion carried out

Reviewer 3

comment

The manuscript is much improved from the previous round of reviews and I commend the authors for taking previous comments on board. I have two main points to raise in this round:

(1) the sample size calculation only works if I assume a higher ICC of around 0.035, not 0.01 as cited in point 3 on page 8, and if I ignore the coefficient of variation of 0.9 cited in point 4, which is much too high given the parameters already specified. I would recommend that the authors drop point 4 as it is determined by other parameters, and also recheck the ICC used in the calculation. Another minor point is to be clear in point 1 on page 7 that the sample size is predicated on an estimated ABSOLUTE reduction of at least 22%

Response

1. A higher ICC of 0.035 was used as suggested

2. Point 4 (Estimate of coefficient of variation of cluster sizes of 0.90) was dropped as suggested

3. Also, the phrase “the sample size is predicted on an estimated absolute reduction of at least 22% of the baseline prevalence”. 

comment

(2) given that the sample size calculation is based on the prevalence of anaemia, it is incorrect to state that the primary outcome is haemoglobin (page 10). The primary outcome should be reported as anaemia, as this is what the sample size calculation is based on. The authors should then go ahead and describe how anaemia will be determined, i.e. measuring haemoglobin using an analyser and classifying children with values below [threshold] as anaemic. Most of this is already indicated in page 10-11. Further, the authors should focus the primary outcome data analysis on anaemia and not haemoglobin (haemoglobin may be analysed as a secondary outcome). If the authors intended haemoglobin to be the primary outcome, then a whole new sample size calculation appropriate for this continuous outcome is required.

Response

Thank you very much for this important correction. The below statement has replaced the previous phrase.

“The primary outcome measure is a reduction in the prevalence of anaemia within the study arms. The study would measure haemoglobin of children using haematological analyser and classify children with Hb <11.0 g/dL as anaemic. Since only children with Hb <11.0 g/dL would be included in the intervention, the baseline prevalence of anaemia in each arm would be 100%. At end line, the prevalence of anaemia within study arm would be determined by estimating the absolute proportion of children whose haemoglobin concentration is <11.0 g/dL.”

Thank you

Benjamin Nuertey

---

## [Editor Report · Decision Letter 3]

16 Mar 2022

Effects of a locally available dietary interventions counselling on the community-based management of anaemia in children under five years in Ghana: Kumbungu cluster randomized controlled trial protocol

PONE-D-20-28388R3

Dear Dr. Nuertey,

I am pleased to inform you that your manuscript has been judged scientifically suitable for publication and will be formally accepted for publication once it meets all outstanding technical requirements.

Kind regards,

Frank Wieringa, M.D., Ph.D.

Academic Editor

PLOS ONE
---

## [Editor Report · Acceptance letter]

13 Apr 2022

PONE-D-20-28388R3 

Effects of a locally available dietary interventions counselling on the community-based management of anaemia in children under five years in Ghana: Kumbungu cluster randomized controlled trial protocol. 

Dear Dr. Nuertey:

I'm pleased to inform you that your manuscript has been deemed suitable for publication in PLOS ONE. Congratulations! Your manuscript is now with our production department. 

Kind regards, 

on behalf of

Dr. Frank Wieringa 

Academic Editor

PLOS ONE